# Purging viral latency by a bifunctional HSV-vectored therapeutic vaccine in chronically SIV-infected macaques

Ziyu Wen[1†], Pingchao Li[2†], Yue Yuan[1†], Congcong Wang[1†], Minchao Li[1], Haohang Wang[1], Minjuan Shi[1], Yizi He[2], Mingting Cui[1], Ling Chen[2*], Caijun Sun[1,3,4,5*]

[1]School of Public Health (Shenzhen), Sun Yat-sen University, Shenzhen, China; [2]State Key Laboratory of Respiratory Disease, Guangzhou Institutes of Biomedicine and Health (GIBH), Chinese Academy of Sciences, Guangzhou, China; [3]Key Laboratory of Tropical Disease Control (Sun Yat-sen University), Ministry of Education, Guangzhou, China; [4]Shenzhen Key Laboratory of Pathogenic Microbes and Biosafety, Shenzhen Campus of Sun Yat-sen University, Shenzhen, China; [5]State Key Laboratory of Anti-Infective Drug Discovery and Development, School of Pharmaceutical Sciences, Sun Yat-sen University, Guangzhou, China

*For correspondence:
chen_ling@gibh.ac.cn (LC);
suncaijun@mail.sysu.edu.cn (CS)

[†]These authors contributed equally to this work

Competing interest: The authors declare that no competing interests exist.

## eLife Assessment

In this **useful** study, the authors tested a novel approach to eradicate the HIV reservoir by constructing a herpes simplex virus (HSV)-based therapeutic vaccine designed to reactivate HIV from latently infected cells and induce an immune response to kill such infected cells. Testing this approach with SIV in a primate model, the authors report that the SIV reservoir was reduced. However, the evidence presented appears to be **incomplete** because the animal group size was small and the SIV reservoir size highly variable.

**Abstract** The persistence of latent viral reservoirs remains the major obstacle to eradicating human immunodeficiency virus (HIV). We herein found that ICP34.5 can act as an antagonistic factor for the reactivation of HIV latency by herpes simplex virus type I (HSV-1), and thus recombinant HSV-1 with ICP34.5 deletion could more effectively reactivate HIV latency than its wild-type counterpart. Mechanistically, HSV-ΔICP34.5 promoted the phosphorylation of HSF1 by decreasing the recruitment of protein phosphatase 1 (PP1α), thus effectively binding to the HIV LTR to reactivate the latent reservoirs. In addition, HSV-ΔICP34.5 enhanced the phosphorylation of IKKα/β through the degradation of IκBα, leading to p65 accumulation in the nucleus to elicit NF-κB pathway-dependent reactivation of HIV latency. Then, we constructed the recombinant HSV-ΔICP34.5 expressing simian immunodeficiency virus (SIV) env, gag, or the fusion antigen sPD1-SIVgag as a therapeutic vaccine, aiming to achieve a functional cure by simultaneously reactivating viral latency and eliciting antigen-specific immune responses. Results showed that these constructs effectively elicited SIV-specific immune responses, reactivated SIV latency, and delayed viral rebound after the interruption of antiretroviral therapy (ART) in chronically SIV-infected rhesus macaques. Collectively, these findings provide insights into the rational design of HSV-vectored therapeutic strategies for pursuing an HIV functional cure.

## Introduction

The epidemic of acquired immunodeficiency syndrome (AIDS), caused by human immunodeficiency virus type I (HIV-1), is still a huge challenge for global public health, with approximately 39.9 million people living with HIV-1 as of 2023. To date, there is neither a curable drug nor a prophylactic vaccine for clinical use against HIV-1 infections (*Sun et al., 2023b*; *Sun et al., 2023a*). Antiretroviral therapy (ART) can effectively control HIV-1 replication to an undetectable level, but the termination of ART usually results in prompt viral rebound from latent viral reservoirs (*Archin et al., 2014*; *Looker et al., 2017*; *Calistri et al., 2003*; *Heng et al., 1994*). Thus, it is of great priority to explore novel strategies for curing HIV latency. The 'shock and kill' strategy is considered a promising approach for purging HIV-1 reservoirs, involving the activation of latently infected cells to express viral products (shock), followed by viral cytopathic effects or specific cytolytic T lymphocytes (CTLs) to eliminate the activated cells (kill) (*Wu et al., 2022*; *Yang et al., 2019*; *Kim et al., 2018*). Numerous latency-reversing agents (LRAs) (*Yang et al., 2019*; *Wu et al., 2022*), including methylation inhibitors, histone deacetylase inhibitors (*Archin et al., 2017*; *Lehrman et al., 2005*), and bromodomain and extra terminal domain protein inhibitors (*Li et al., 2013*; *Bisgrove et al., 2007*), have been identified to reactivate latent HIV-1 in preclinical studies, but there is no ideal LRA available for clinical patients yet.

Herpes simplex virus (HSV), a human herpesvirus, features a 152 kb double-stranded DNA genome encoding over 80 proteins (*Poh, 2016*). Owing to its distinctive genetic background, high capacity, broad tropism, thermostability, and excellent safety profile, the modified HSV constructs have extensive applications in gene therapy and oncolytic virotherapy. For example, talimogene laherparepvec (T-VEC), an HSV-1 variant with ICP34.5 deletion and GM-CSF insertion, received FDA approval in 2015 for treating malignant melanoma, showcasing notable safety and efficacy in clinical practice. Recombinant HSV-based constructs have also emerged as efficacious gene delivery vectors against infectious diseases. Early studies indicated that prophylactic vaccines based on HSV, expressing simian immunodeficiency virus (SIV) antigens, could elicit robust antigen-specific CTL responses in mice and monkeys, providing enduring and partial protection against pathogenic SIVmac239 challenges (*Kaur et al., 2007*; *Murphy et al., 2000*). Moreover, increasing data suggest the crucial role of HIV-specific CTL in controlling viral replication and eliminating potential HIV reservoirs (*Collins et al., 2020*; *Leitman et al., 2017*). Significantly, epidemiological research suggests a synergistic effect between HSV and HIV infections, with HSV infection in HIV patients being associated with increased HIV-1 viral load and disease progression (*Looker et al., 2017*; *Calistri et al., 2003*; *Heng et al., 1994*). Some studies have further unveiled the ability of HSV to activate HIV latent reservoirs (*Amici et al., 2004*; *Amici et al., 2001*; *Pierce et al., 2023*). Given the potential of HSV to simultaneously induce antigen-specific immune responses and reactivate latent viral reservoirs, we propose a proof-of-concept strategy to achieve an HIV functional cure using a modified bifunctional HSV-vectored therapeutic vaccine.

## Results

### The modified HSV-ΔICP34.5-based constructs reactivated HIV latency more efficiently than wild-type HSV counterparts

The J-Lat 10.6 cells, derived from Jurkat T cells containing latent HIV-1 provirus, were infected with wild-type HSV-1 Mckrae strain at different multiplicities of infection (MOIs) to assess its capability to reactivate HIV latency. Flow cytometry analysis showed a dose-dependent increase in green fluorescent protein (GFP) expression (*Figure 1A*), indicating that HSV-1 can reactivate latent HIV. Subsequently, we infected J-Lat 10.6 cells with an HSV-1 17 strain containing GFP (HSV-GFP) and observed a significant upregulation in the mRNA levels of HIV-1-driven transcripts, including Tat, Gag, Vif, and Vpr, along with 5'-LTR-containing RNA. This demonstrates that the attenuated HSV-1 strain can reactivate latent HIV at different MOIs (*Figure 1B*). Using the bacterial artificial chromosome (BAC)/galactokinase (galK) system, we then constructed the recombinant HSV deleted with ICP34.5 (HSV-ΔICP34.5) based on HSV-GFP. Notably, we discovered for the first time that the HSV-ΔICP34.5 reactivated HIV latency more efficiently than its parental strain (HSV-GFP). Specifically, the mRNA levels of HIV-1 transcripts, including those driven by the 5' LTR, as well as Tat, Gag, Vpr, Vif were significantly higher in J-Lat 10.6 cells infected with HSV-ΔICP34.5 compared to those treated with HSV-GFP (*Figure 1C*), despite the weaker replication ability of HSV-ΔICP34.5 in these cells, as indicated by the mRNA level of HSV UL27 (*Figure 1D*, *Figure 1—figure supplement 1*).

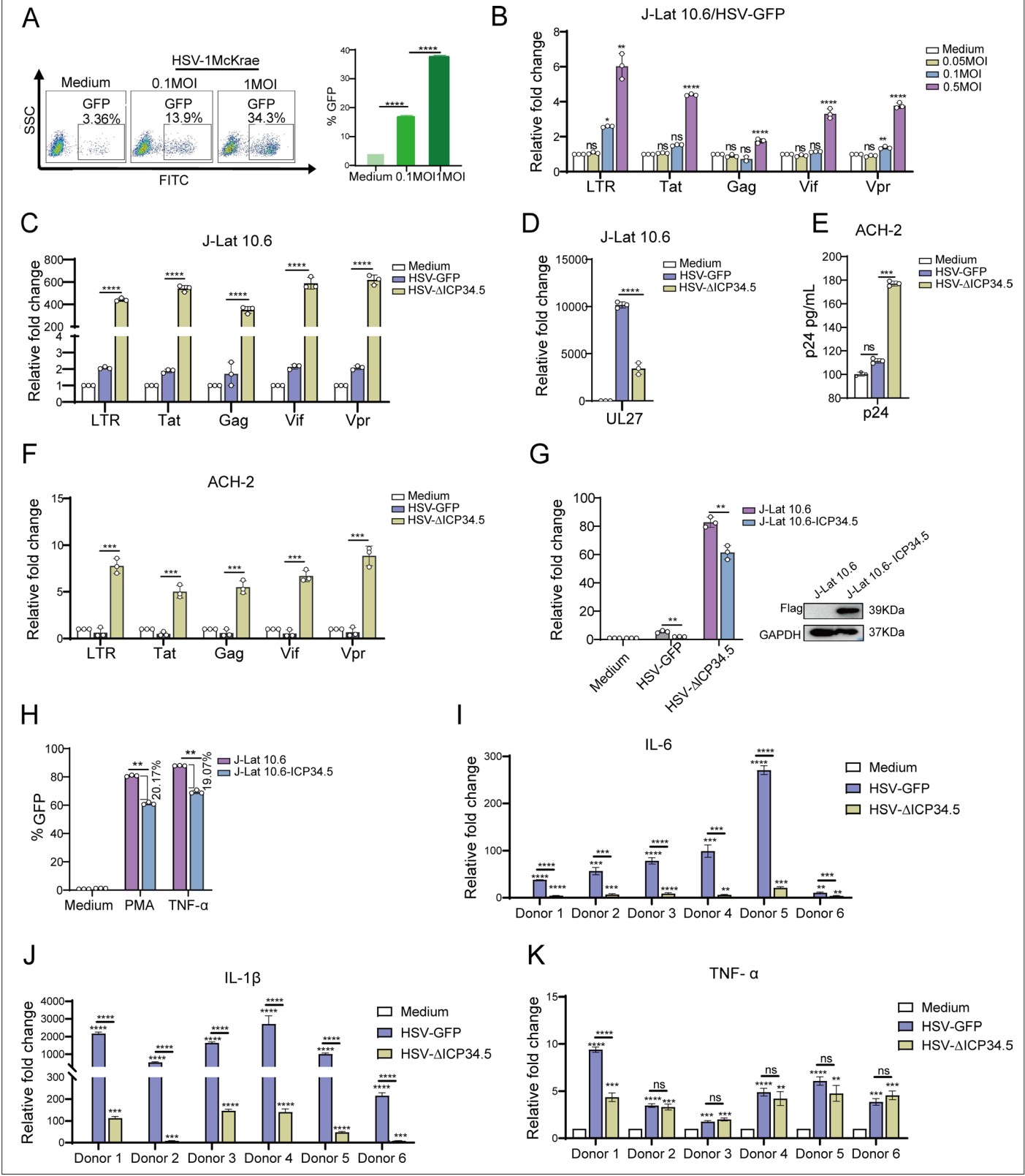

**Figure 1.** The modified HSV-ΔICP34.5-based constructs reactivated human immunodeficiency virus (HIV) latency more efficiently than wild-type herpes simplex virus (HSV) counterparts. (**A**) J-Lat 10.6 cells (1×10⁶) were infected with varying multiplicities of infection (MOIs) of wild-type HSV type I (HSV-1) Mckrae strain for 30 hr. The proportion of green fluorescent protein (GFP⁺) cells, indicating activated latent cells, is shown in the pseudocolor plot (left) and the corresponding bar chart (right). (**B**) J-Lat 10.6 cells (1×10⁶) were infected with varying MOIs of HSV-1 17 strain containing GFP (HSV-GFP) for

*Figure 1 continued on next page*

*Figure 1 continued*

30 hr, and the mRNA levels of HIV-1 transcripts driven by the 5′ LTR, as well as Tat, Gag, Vpr, and Vif, are presented in the histogram. (**C**) J-Lat 10.6 cells (1×10⁶) were infected with HSV-GFP or HSV-ΔICP34.5 at an MOI of 0.1 for 30 hr. The mRNA levels of HIV-1 transcripts driven by the 5′ LTR, as well as Tat, Gag, Vpr, Vif, and (**D**) HSV-1 UL27 are presented in the histogram. (**E**) ACH-2 cells (1×10⁶) were infected with HSV-GFP or HSV-ΔICP34.5 at an MOI of 0.1 for 30 hr. The p24 protein level was detected using an HIV-1 p24 ELISA kit, and the mRNA levels of HIV-1 transcripts driven by the 5′ LTR, as well as Tat, Gag, Vpr, and Vif, are shown in the histogram (**F**). (**G**) J-Lat 10.6 and J-Lat 10.6-ICP34.5 cells were infected with HSV-GFP or HSV-ΔICP34.5, and the mRNA levels of HIV-1 Tat were shown with the histogram (left). Blotting showed that J-Lat 10.6 cells stably expressing HSV ICP34.5 (J-Lat 10.6-ICP34.5) can appropriately express ICP34.5 protein using Flag-tag antibodies (right). (**H**) J-Lat 10.6 and J-Lat 10.6-ICP34.5 cells were respectively stimulated with phorbol 12-myristate 13-acetate (PMA) (10 ng/mL) and TNF-α (10 ng/mL), and the expression level of GFP⁺ cells is displayed with the corresponding bar chart. (**I–K**) Primary CD4⁺ T cells from people living with HIV (PLWH) were infected with HSV-GFP or HSV-ΔICP34.5. The inflammatory response was assessed by evaluating mRNA levels of IL-6, IL-1β, and TNF-α using qPCR. Data shown are mean ± SD. Three independent experiments were repeated. **p<0.01, ***p<0.001, ****p<0.0001. ns: no significance.

The online version of this article includes the following source data and figure supplement(s) for figure 1:

**Source data 1.** PDF file containing original western blots for *Figure 1G*, indicating the relevant bands and treatments.

**Source data 2.** Original files for western blot analysis displayed in *Figure 1G*.

**Source data 3.** Source data for *Figure 1A–K*.

**Figure supplement 1.** Viral growth curve.

**Figure supplement 2.** The reactivation effect of herpes simplex virus type I (HSV-1) on the latent human immunodeficiency virus (HIV) reservoir in primary CD4⁺ T cells from people living with HIV (PLWH).

**Figure supplement 3.** Adenovirus and vaccinia virus cannot reactivate human immunodeficiency virus (HIV) latency in J-Lat 10.6 cells.

**Figure supplement 4.** Deletion of ICP0 diminished the reactivation effect of human immunodeficiency virus (HIV) latency by herpes simplex virus type I (HSV-1).

**Figure supplement 5.** ICP0 promotes human immunodeficiency virus (HIV) reactivation in J-Lat10.6 cells.

Furthermore, this finding was validated in ACH-2 cells, which are derived from T cells latently infected with replication-competent HIV-1. A significantly higher level of p24 protein, a key indicator of HIV replication, was found in the HSV-ΔICP34.5-infected ACH-2 cells compared to HSV-GFP-treated cells (*Figure 1E*). Additionally, the mRNA levels of HIV-related transcripts, including those driven by the 5′ LTR, as well as Tat, Gag, Vpr, Vif, were also significantly increased (*Figure 1F*). Subsequently, we generated J-Lat 10.6 cells stably expressing ICP34.5-Flag-Tag (J-Lat 10.6-ICP34.5) using a recombinant lentivirus system and confirmed the expression of the ICP34.5 protein. Our research revealed that HSV-ΔICP34.5 displayed a reduced capacity to reverse HIV latency in J-Lat 10.6-ICP34.5 cells compared to J-Lat 10.6 cells (*Figure 1G*). Moreover, in J-Lat 10.6-ICP34.5 cells, the potency of latent reversal agents such as phorbol 12-myristate 13-acetate (PMA) and TNF-α was notably reduced when contrasted with J-Lat 10.6 cells (*Figure 1H*). We also confirmed the enhanced reactivation of HIV latency by HSV-ΔICP34.5 in primary CD4⁺ T cells from people living with HIV (PLWH) (*Figure 1—figure supplement 2*). In addition, HSV-ΔICP34.5 induced a lower level of inflammatory cytokines (including IL-6, IL-1β, and TNF-α) in primary CD4⁺ T cells from PLWH compared to HSV-GFP stimulation, likely due to its lower virulence and replication ability (*Figure 1I–K*). Thus, these data suggested that the safety of HSV-ΔICP34.5 in PLWH might be tolerable, at least in vitro. In addition, neither adenovirus nor vaccinia virus was able to reactivate HIV latency in this study (*Figure 1—figure supplement 3*), and the deletion of ICP0 gene from HSV-1 diminished its reactivation effect on HIV latency (*Figure 1—figure supplement 4*). Importantly, overexpressing ICP0 greatly reactivates latent HIV (*Figure 1—figure supplement 5*), implying that this reactivation is virus-specific and that ICP0 plays an important role in reversing HIV latency. Overall, these findings indicated that the recombinant HSV-ΔICP34.5 construct specifically reactivated HIV latency with high efficiency in the presence of ICP0.

## The modified HSV-based constructs effectively reactivated HIV latency by modulating the IKKα/β-NF-κB pathway and PP1-HSF1 pathway

Next, based on RNA-seq analysis for HSV-ΔICP34.5-induced signaling pathways (*Figure 2—figure supplement 1*) and previous literature, we explored the mechanism of reactivating viral latency by HSV-ΔICP34.5-based constructs. J-Lat 10.6 cells were infected with HSV-GFP or HSV-ΔICP34.5, and results showed that HSV-ΔICP34.5 significantly enhanced the phosphorylation of IKKα/β, promoted the degradation of IKBα, and thus led to the accumulation of p65 in the nucleus (*Figure 2A*). NF-κB is

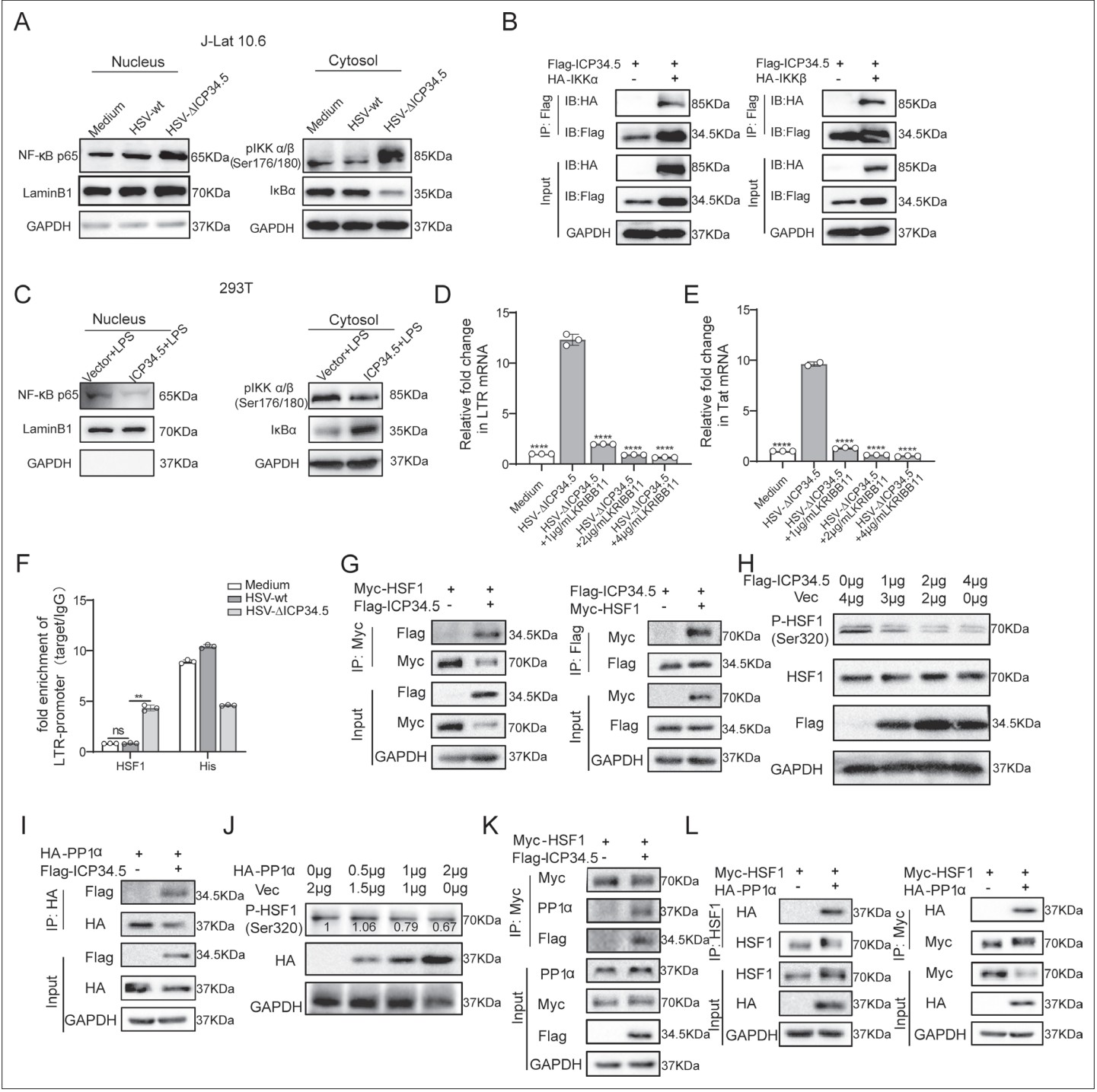

**Figure 2.** The modified herpes simplex virus (HSV)-based constructs effectively reactivated human immunodeficiency virus (HIV) latency by modulating the NF-κB pathway and HSF1 pathway. (**A**) J-Lat 10.6 cells were infected with HSV-green fluorescent protein (GFP) and HSV-ΔICP34.5 at a multiplicity of infection (MOI) of 0.1. Cytoplasmic and nuclear protein fractions were analyzed for NF-κB p65, phosphorylated IKKα/β (p-IKKα/β), and IκBα. GAPDH and Lamin B1 served as loading controls for cytoplasmic and nuclear fractions, respectively. (**B**) 293T cells were co-transfected with Flag-ICP34.5 and either IKKα (left) or IKKβ (right), and the interactions were examined by co-immunoprecipitation (Co-IP) assays. (**C**) 293T cells were transfected with Flag-ICP34.5 or an empty vector (Vec) for 24 hr and then treated with lipopolysaccharide (LPS; 1 μg/mL) for 8 hr. Cytoplasmic and nuclear protein fractions were subsequently analyzed by Western blot (WB). (**D–E**) J-Lat 10.6 cells were infected with HSV-ΔICP34.5 and treated with increasing concentrations of KRIBB11 concentrations. The mRNA levels of Tat and of HIV-1 transcripts driven by the 5' LTR were quantified by qPCR. (**F**) J-Lat 10.6 cells were infected with HSV-wt or HSV-ΔICP34.5 at an MOI of 0.1 for 36 hr. Chromatin immunoprecipitation followed by qPCR (ChIP–qPCR) was performed to assess HSF1 binding to LTR. Normal IgG and an anti-histone H3 antibody served as negative and positive controls, respectively. (**G**) 293T cells were co-transfected

*Figure 2 continued on next page*

*Figure 2 continued*

with Flag-ICP34.5 and Myc-HSF1, and the interactions were examined by Co-IP assays. (**H**) The immunoblot depicts the alterations in protein levels in 293T cells transfected with either empty vector or ICP34.5 for 6 hr, followed by treatment with 10 μM MG132 for 24 h. (**I**) 293T cells transfected with Flag-ICP34.5 and HA-PP1α, analyzed by Co-IP assays. (**J**) The immunoblot depicts the alterations in protein levels in 293T cells transfected with either empty vector or HA-PP1α for 6 hr, followed by 24 hr of treatment with 10 μM MG132. (**K**) 293T cells transfected with Myc-HSF1 with or without Flag-ICP34.5, analyzed by Co-IP assays. (**L**) 293T cells transfected with Myc-HSF1 and HA-PP1α, analyzed through Co-IP assays. Data shown are mean ± SD. **p<0.01, ****p<0.0001. ns: no significance.

The online version of this article includes the following source data and figure supplement(s) for figure 2:

**Source data 1.** PDF file containing original western blots for *Figure 2A–C and G–L*, indicating the relevant bands and treatments.

**Source data 2.** Original files for western blot analysis displayed in *Figure 2A–C and G–L*.

**Source data 3.** Source data for *Figure 2D–F*.

**Figure supplement 1.** RNA-seq analysis for HSV-ΔICP34.5-induced signaling pathways.

**Figure supplement 2.** The overexpression of ICP34.5 have no influence on the level of HSF1 expression.

a well-known host transcription factor that exists in the form of the NF-κB-IκB complex in resting cells, but IκB can degrade and release the NF-κB dimer to enter the nucleus and promote gene transcription in response to external stimulation. Using the co-immunoprecipitation (Co-IP) assay, we verified that the ICP34.5 protein had a specific interaction with IKKα/β, and then ICP34.5 could dephosphorylate IKKα/β. Moreover, the overexpression of ICP34.5 effectively inhibited lipopolysaccharide-induced NF-κB pathway activation by inhibiting p65 entry into the nucleus (*Figure 2B and C*).

To further clarify the underlying mechanism, we performed the immunoprecipitation-mass spectrometry analysis in ICP34.5 overexpressing cells to identify other potential molecules contributing to this reactivation, and we found that ICP34.5 can also interact with heat shock 1 protein (HSF1) (*Supplementary file 1a*). HSF1 has been reported as a transcription factor correlated with the reactivation of HIV latency (*Xu et al., 2022*; *Lin et al., 2018*; *Zeng et al., 2017*). To test whether HSF1 contributes to the reactivation of HIV latency by HSV-ΔICP34.5-based constructs, KRIBB11, an inhibitor of HSF1, was administered to HSV-ΔICP34.5-infected J-Lat 10.6 cells. The results indicated that the reactivation ability of HSV-ΔICP34.5 was significantly inhibited by KRIBB11 treatment in a dose-dependent manner (*Figure 2D and E*). Furthermore, a significant enhancement of the binding of HSF1 to the HIV LTR was observed upon HSV-ΔICP34.5 infection, leading to an increase in the reactivation of HIV latency (*Figure 2F*). The direct interaction between ICP34.5 and HSF1 was also identified by Co-IP assay. Importantly, HSF1 was effectively dephosphorylated at Ser320 as a result of the overexpression of ICP34.5, while no influence on the level of HSF1 expression was observed (*Figure 2G and H*, *Figure 2—figure supplement 2*). Considering that protein phosphatase 1 (PP1) can interact with ICP34.5 and dephosphorylate eIF2α (*Li et al., 2011*), we then investigated the interaction between PP1α and ICP34.5 (*Figure 2I*). Additionally, a direct interaction between PP1α and HSF1 was found, allowing for the dephosphorylation of HSF1 and then affecting its ability to reactivate HIV latency (*Figure 2J–L*). Collectively, these findings demonstrated that our modified HSV-ΔICP34.5-based constructs effectively reactivated HIV latency by modulating the IKKα/β-NF-κB pathway and PP1-HSF1 pathway.

## Recombinant HSV-ΔICP34.5 constructs expressing SIV antigens elicited robust immune responses in mice

Subsequently, we explored the potential of HSV-ΔICP34.5 as a bifunctional therapeutic vector to not only reactivate latent viral reservoirs but also induce antigen-specific immune responses against viral replication. To achieve this, SIV antigen genes were introduced into HSV-ΔICP34.5 vector based on BAC/galK system. Additionally, the ICP47 gene was deleted to augment the immunogenicity of the HSV vector (*Figure 3A*, *Figure 3—figure supplement 1*). A series of recombinant HSV-ΔICP34.5ΔICP47-based vectors expressing SIV gag and env antigen were constructed, and the antigen expression of these constructs was confirmed by western blotting assay (*Figure 3B and C*). Furthermore, to improve the immunogenicity of the targeted antigen, we fused the SIV gag with soluble PD1 (sPD1), enabling it to competitively bind with PD-L1 and thereby block the PD1/PD-L1 immune inhibitory pathway (*Figure 3D*). Consistent with the above findings, these HSV-ΔICP34.5ΔICP47-based SIV vaccines also efficiently reactivated HIV latency (*Figure 3—figure supplement 2*). Then, the immunogenicity of

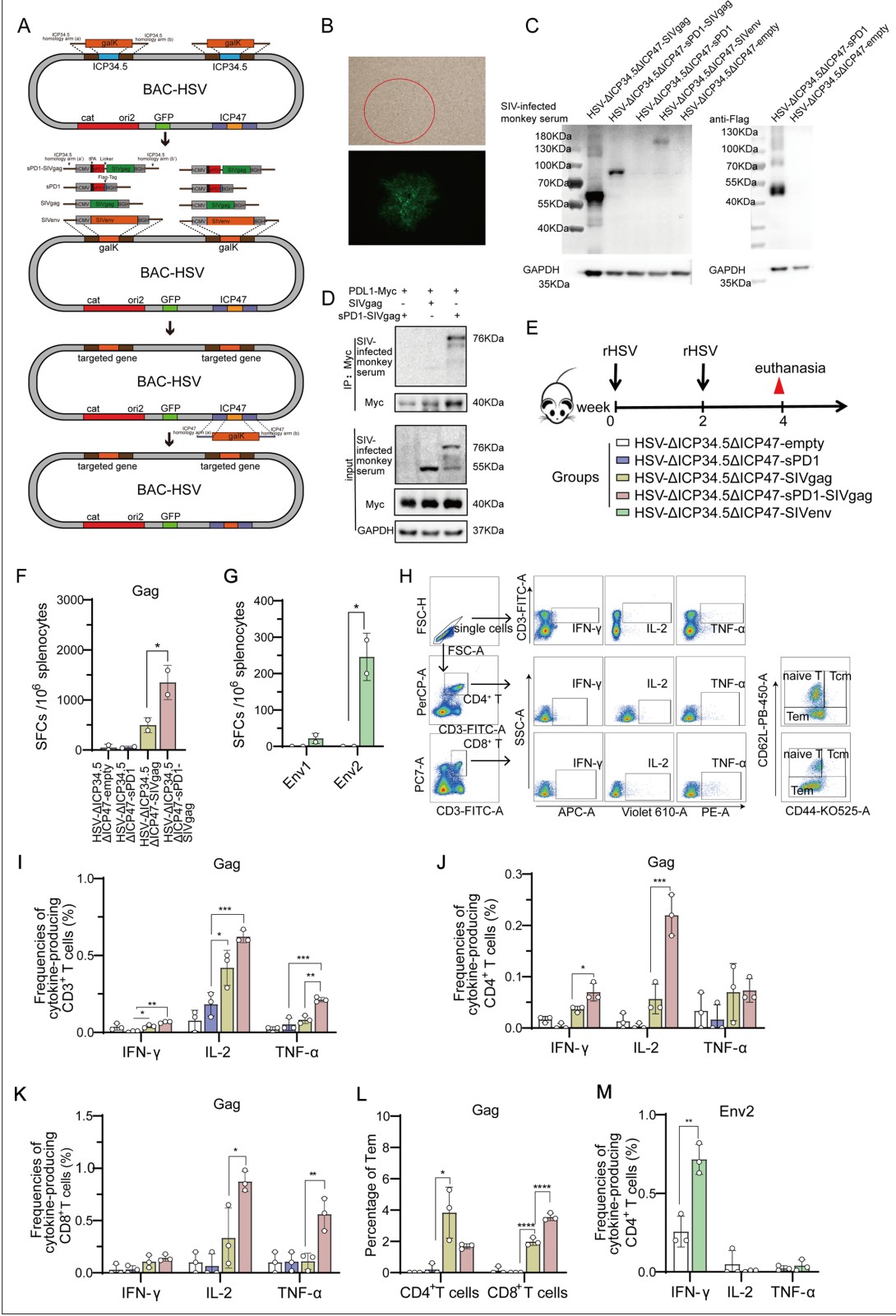

**Figure 3.** Recombinant herpes simplex virus type I (HSV-1) vector-based simian immunodeficiency virus (SIV) vaccines induce specific T cell immune responses in mice. (**A**) Schematic diagram illustrating the construction of recombinant HSV through the bacterial artificial chromosome (BAC)/galactokinase (galK) selection system. The ICP34.5 gene was replaced with the galK gene via homologous recombination, followed by substituting galK with a target gene expression cassette containing the hCMV promoter and BGH terminator. Finally, the ICP47 gene was deleted. (**B**) Brightfield

*Figure 3 continued*

(top) and fluorescence (bottom) images of a clone of the rescued recombinant HSV. (**C**) Vero cells infected with recombinant HSV constructs, with protein expression of targeted genes detected using SIV-infected monkey serum. (**D**) HeLa cells transfected with Myc-PDL1 and then infected with HSV-ΔICP34.5ΔICP47-empty, HSV-ΔICP34.5ΔICP47-SIVgag, or HSV-ΔICP34.5ΔICP47-sPD1-SIVgag at a multiplicity of infection (MOI) of 0.1 for 24 hr. Cell lysates were subjected to co-immunoprecipitation (Co-IP) analysis. (**E**) Schematic schedule of mouse vaccination. Twenty-five mice were randomly allocated to five groups: HSV-ΔICP34.5ΔICP47-empty, HSV-ΔICP34.5ΔICP47-sPD1, HSV-ΔICP34.5ΔICP47-SIVgag, HSV-ΔICP34.5ΔICP47-sPD1-SIVgag, and HSV-ΔICP34.5ΔICP47-SIVenv. Mice were injected with the corresponding vaccines at weeks 0 and 2. At week 4, mice were sacrificed, and spleen lymphocytes were collected to evaluate immune response.(**F–G**) Column graphs showing the number of Gag or Env1, Env2-specific spot-forming cells (SFCs) per $10^6$ spleen lymphocytes, as measured by interferon γ (IFN-γ) ELISpot assay. (**H**) Pseudocolor plot of flow cytometry illustrating the gating strategy. Column graphs showing the frequencies of IFN-γ, IL-2, and TNF-α production from gag-specific CD3$^+$ T (**I**), CD4$^+$ T (**J**), and CD8$^+$ T cells (**K**). (**L**) Bar chart showing the proportion of Tem (effector memory T cells) among CD4$^+$ T and CD8$^+$ T cells upon stimulation with the SIV Gag peptide pools.(**M**) Bar chart showing the frequencies of Env2-specific IFN-γ$^+$ CD4$^+$ T cells. Data were expressed as mean ± SD from five mice samples. Three independent experiments for the animal immunization were repeated. *p<0.05, **p<0.01, ***p<0.001, ****p<0.0001. ns: no significance.

The online version of this article includes the following source data and figure supplement(s) for figure 3:

**Source data 1.** PDF file containing original western blots for *Figure 3C*, indicating the relevant bands and treatments.

**Source data 2.** Original files for western blot analysis displayed in *Figure 3C*.

**Source data 3.** Source data for *Figure 3F–G and I–M*.

**Figure supplement 1.** The identification of recombinant pBAC-HSV plasmid.

**Figure supplement 2.** HSV-ΔICP34.5ΔICP47-based recombinant viruses reverse HIV latency in J-Lat 10.6 cells.

the above modified HSV-ΔICP34.5ΔICP47-based SIV vaccine was assessed in mice (*Figure 3E*). Our results showed that these constructs effectively elicited SIV antigen-specific T cell immune responses using the interferon γ (IFN-γ) ELISpot assay and the intracellular cytokine staining (ICS) assay. Of note, the frequency of SIV Gag-specific IFN-γ -secreting spot-forming cells (SFCs) in the HSV-sPD1-SIVgag group (1350 SFCs per $10^6$ splenocytes) was significantly higher than that in the HSV-SIVgag group (498 SFCs per $10^6$ splenocytes) (*Figure 3F*). The frequency of SIV Env2-specific IFN-γ -secreting SFCs in the HSV- SIVenv group was significantly higher than the HSV-empty group (*Figure 3G*). Furthermore, the polyfunctionality of antigen-specific T cell subsets in response to SIV antigen stimulation was confirmed using the ICS assay (*Figure 3H–M*). Consistently, the HSV-sPD1-SIVgag group showed a significantly higher frequency of SIV-specific CD3$^+$ T, CD4$^+$ T, and CD8$^+$ T cell subsets secreting IFN-γ, IL-2, and TNF-α cytokines compared to the HSV-SIVgag group (*Figure 3I–K*). Notably, a heightened proportion of Gag-specific effector memory T cells (Tem) of CD8$^+$ T cell subset in HSV-ΔICP34.5ΔICP47-sPD1-SIVgag group was observed in comparison to the HSV-Gag group (*Figure 3L*). In addition, a higher frequency of SIV- Env2-specific CD4$^+$ T cells secreting IFN-γ was observed in the HSV-ΔICP34.5ΔICP47-SIVenv group than in the HSV-ΔICP34.5ΔICP47-empty group (*Figure 3M*). These data indicated that the vaccines constructed in this study elicited a robust T cell immune response in mice. Moreover, the blockade of PD1/PDL1 signaling pathway effectively enhanced vaccine-induced T cell immune responses, which was consistent with our and other previous studies (*Zhou et al., 2013*; *Wu et al., 2022*; *Pan et al., 2018*; *Xiao et al., 2014*).

## The modified HSV-based constructs efficiently elicited SIV-specific immune responses in chronically SIV-infected macaques

Next, the immunogenicity of these HSV-ΔICP34.5ΔICP47-vectored SIV vaccines was further investigated in chronically SIV$_{mac239}$-infected rhesus macaques (RMs). To mimic chronically infected HIV patients in clinic practice, all RMs used in this study were chronically infected with SIV and received ART treatment for several years, as reported in our previous studies (*Pan et al., 2018*; *Yang et al., 2019*; *Wu et al., 2021*; *Wu et al., 2022*; *He et al., 2023*). Based on sex, age, viral load, and CD4 count, nine RMs were assigned into three groups: ART+saline group (n=3), ART+HSV-ΔICP34.5ΔICP47-empty group (n=3), and ART+HSV-ΔICP34.5ΔICP47-sPD1-SIVgag/SIVenv group (n=3) (*Supplementary file 1b*). All RMs received ART (FTC/6.7 mg/kg/once daily, PMPA/10 mg/kg/once daily) treatment to avoid the interference of free SIV particles. On day 33 and day 52, these RMs were immunized with saline, HSV-ΔICP34.5ΔICP47-empty, or HSV-ΔICP34.5ΔICP47-sPD1-SIVgag/SIVenv respectively. On day 70, ART treatment in all RMs was discontinued to evaluate the time interval of viral rebound. Samples were collected at different time points to monitor virological and immunological parameters

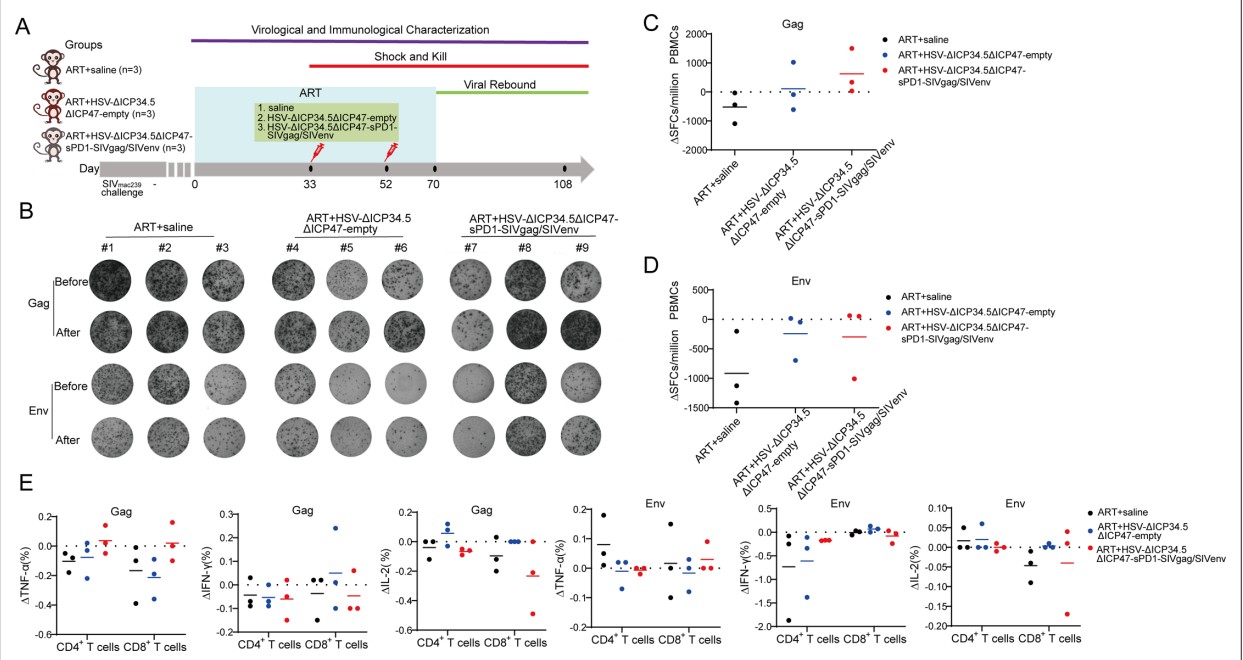

**Figure 4.** The modified herpes simplex virus (HSV)-based constructs efficiently elicited simian immunodeficiency virus (SIV)-specific immune responses in chronically SIV-infected macaques. (**A**) Schematic schedule of the macaque experiment. Nine chronically SIV-infected macaques were assigned into three groups: antiretroviral therapy (ART)+saline group (n=3), ART+HSV-ΔICP34.5ΔICP47-empty group (n=3), and ART+HSV-ΔICP34.5ΔICP47-sPD1-SIVgag/SIVenv group (n=3). All SIV-infected macaques received ART treatment (FTC/6.7 mg/kg/once daily, PMPA/10 mg/kg/once daily) for 33 days. On days 33 and 52, macaques were immunized with saline, HSV-ΔICP34.5ΔICP47-empty, and HSV-ΔICP34.5ΔICP47-sPD1-SIVgag/SIVenv respectively. ART treatment,was interrupted in all macaques on day 70 after the second vaccination. Samples were collected at various time points to monitor virological and immunological parameters. (**B**) Representative images of Gag or Env-specific spots (2.5×10^5 cells per well) from each macaque pre-vaccination (before, day 33) and post-vaccination (after, day 70) by ELISpot assay. (**C–D**) Difference in SIV-specific interferon γ (IFN-γ)-secreting cells (ΔSFCs) between pre- and post-immunization, assessing the immune response induced by HSV-ΔICP34.5ΔICP47-vectored SIV vaccines. (**E**) Difference in SIV-specific TNF-α/IFN-γ/IL-2-secreting CD4+ T and CD8+ T subsets between pre- and post-immunization, detected by intracellular cytokine staining (ICS) assay.

The online version of this article includes the following source data for figure 4:

**Source data 1.** Source data for *Figure 4C–E*.

(*Figure 4A*). To reduce the impact of individual RM variations, the difference in SIV-specific IFN-γ-secreting SFCs between post-immunization and pre-immunization (ΔSFCs) was used to evaluate the immune response induced by HSV-ΔICP34.5ΔICP47-vectored SIV vaccines. The results showed that SIV Gag-specific ΔSFCs in the ART+HSV-ΔICP34.5ΔICP47-sPD1-SIVgag/SIVenv group were greatly increased when compared with those in the ART+HSV-ΔICP34.5ΔICP47-empty group and ART+saline group (*Figure 4B–D*). A similar enhancement of SIV Gag-specific TNF-α -secreting CD4+ T and CD8+ T subsets was also verified by ICS assay (*Figure 4E*). Collectively, these data demonstrated that the HSV-ΔICP34.5ΔICP47-sPD1-SIVgag/SIVenv construct elicited robust SIV-specific T cell immune responses in ART-treated, SIV-infected RMs.

## The modified HSV-based constructs partly reactivated SIV latency in vivo and delayed viral rebound in chronically SIV-infected, ART-treated macaques

Finally, we investigated the therapeutic efficacy of HSV-ΔICP34.5ΔICP47-sPD1-SIVgag/SIVenv in chronically SIV-infected, ART-treated RMs. Consistent with our previous studies, the plasma viral load (VL) in these RMs was effectively suppressed during ART treatment, but rebounded after ART discontinuation. The VL in the ART+saline group promptly rebounded after ART discontinuation, and one of three RMs showed a significantly increased peak VL compared with the pre-ART VL (*Figure 5A and D*). However, plasma VL in the ART+HSV-ΔICP34.5ΔICP47-empty group and the ART+HSV-ΔICP34.5ΔICP47-sPD1-SIVgag/SIVenv group exhibited a delayed rebound interval (*Figure 5B and C*).

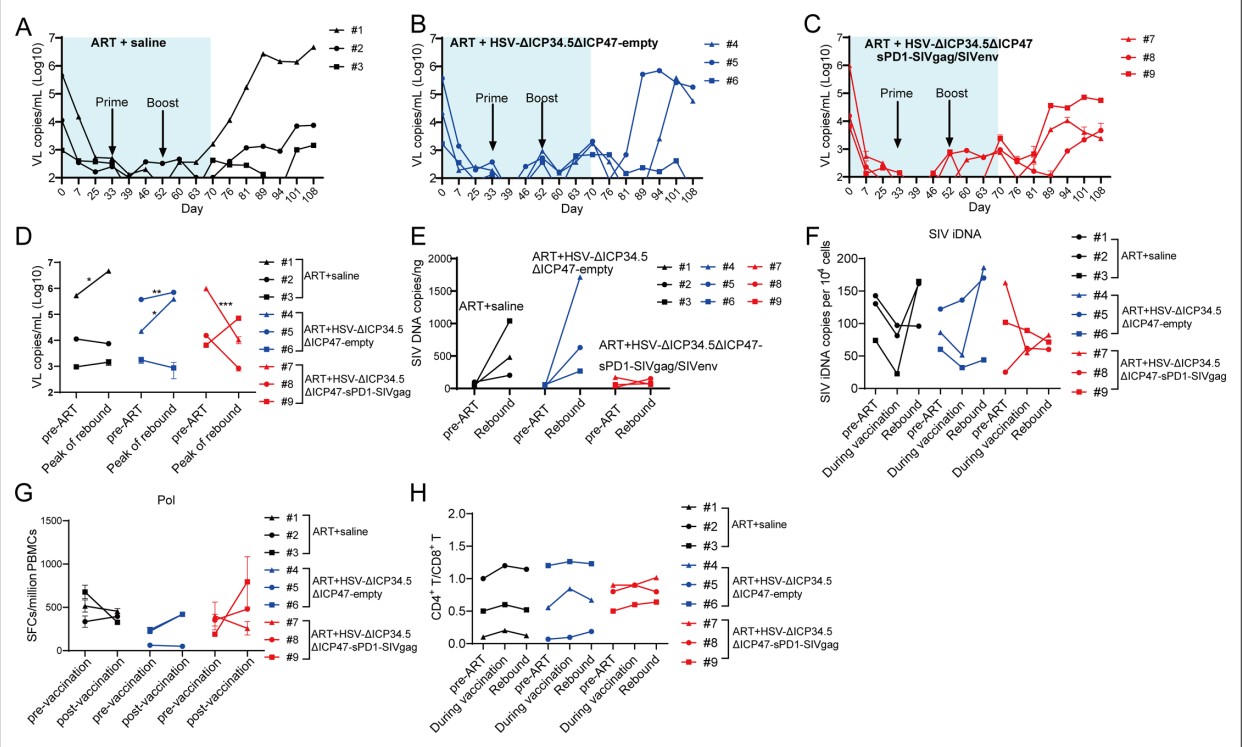

**Figure 5.** The modified herpes simplex virus (HSV)-based constructs effectively reactivated simian immunodeficiency virus (SIV) latency in vivo in chronically SIV-infected, antiretroviral therapy (ART)-treated macaques. (**A–C**) Viral load (VL) changes in plasma for each animal were monitored throughout the experiment using real-time PCR. The detection limit is 100 copies/mL plasma. The shaded area represented the duration of ART administration. (**D**) The VL change in plasma between pre-ART and the peak value in the rebound stage after ART discontinuation. (**E**) Change in total SIV DNA copies between pre-ART and viral rebound after ART discontinuation. (**F**) SIV integrated DNA (iDNA) copy numbers detected by Alu-PCR at different time points. (**G**) Change in the number of SIV Pol-specific IFN-γ-secreting cells between pre-immunization (day 33) and post-immunization (day 70), as detected by ELISpot assay. (**H**) Change in the CD4$^+$ T/ CD8$^+$ T ratio. Data shown are mean ± SD. **p<0.01, ***p<0.0001, ****p<0.0001.

The online version of this article includes the following source data and figure supplement(s) for figure 5:

**Source data 1.** Source data for *Figure 5A–H*.

**Figure supplement 1.** Longitudinal body-weight profiles of macaques throughout the study period.

**Figure supplement 2.** Effect on the cell composition of peripheral blood in the macaques of different groups.

Remarkably, there was a lower rebounded peak VL (two of three RMs) than pre-ART VL in the ART+HSV-ΔICP34.5ΔICP47-sPD1-SIVgag/SIVenv group, while two of three RMs had a higher rebounded peak VL than pre-ART VL in the ART+HSV-ΔICP34.5ΔICP47-empty group (*Figure 5D*). Then, we assessed the potential effect on the latent SIV reservoirs in vivo by administering our modified HSV-based SIV therapeutic constructs in these RMs. Although there were no obvious viral blips observed in these RMs, we found significant suppression of total SIV DNA and integrated SIV DNA provirus in the ART+HSV-ΔICP34.5ΔICP47-sPD1-SIVgag/SIVenv group. However, the copies of the SIV DNA provirus were significantly improved in the ART+HSV-ΔICP34.5ΔICP47-empty group and ART+saline group (*Figure 5E and F*). More interestingly, we assessed the magnitude of SIV Pol antigen-specific immune responses, which could represent to some extent the level of newly generated virions from the reactivated SIV reservoirs, because SIV Pol antigen was not included in our designed vaccine constructs. Specifically, the Pol-specific SFCs amount on day 70 (511 SFCs/10$^6$ peripheral blood mononuclear cells [PBMCs], post-vaccination) was higher than that on day 33 (315 SFCs/10$^6$ PBMCs, pre-vaccination) in the ART+HSV-ΔICP34.5ΔICP47-sPD1-SIVgag/SIVenv group. In addition, there was a similar observation in the ART+HSV-ΔICP34.5ΔICP47-empty group. In contrast, the Pol-specific SFCs gradually decreased with the duration of ART treatment in the ART+saline group (*Figure 5G*). In addition, the CD4$^+$/CD8$^+$ T cell ratio (*Figure 5H*) and body weight (*Figure 5—figure supplement 1*) after treatment were effectively ameliorated in the RMs of the ART+HSV-ΔICP34.5ΔICP47-sPD1-SIVgag/SIVenv group, but not in the ART+HSV-ΔICP34.5ΔICP47-empty group and ART+saline group. Our data also

demonstrated that there was no significant effect on the cell composition of peripheral blood in the macaques of ART+HSV-ΔICP34.5ΔICP47-sPD1-SIVgag/SIVenv group (*Figure 5—figure supplement 2*). Taken together, these findings suggested that the latent SIV reservoirs might be effectively purged because of the effect of simultaneously reactivating viral latency and eliciting SIV-specific immune responses by our modified HSV-based SIV therapeutic constructs, thus resulting in a delayed viral rebound in chronically SIV-infected, ART-treated macaques.

## Discussion

To conquer the continuous epidemic of AIDS, exploring novel strategies to render and eliminate HIV latency should stand as a pivotal pursuit. Currently, numerous strategies, including shock and kill, block and lock, chimeric antigen receptor T cell therapy, therapeutic vaccination, and gene editing, have

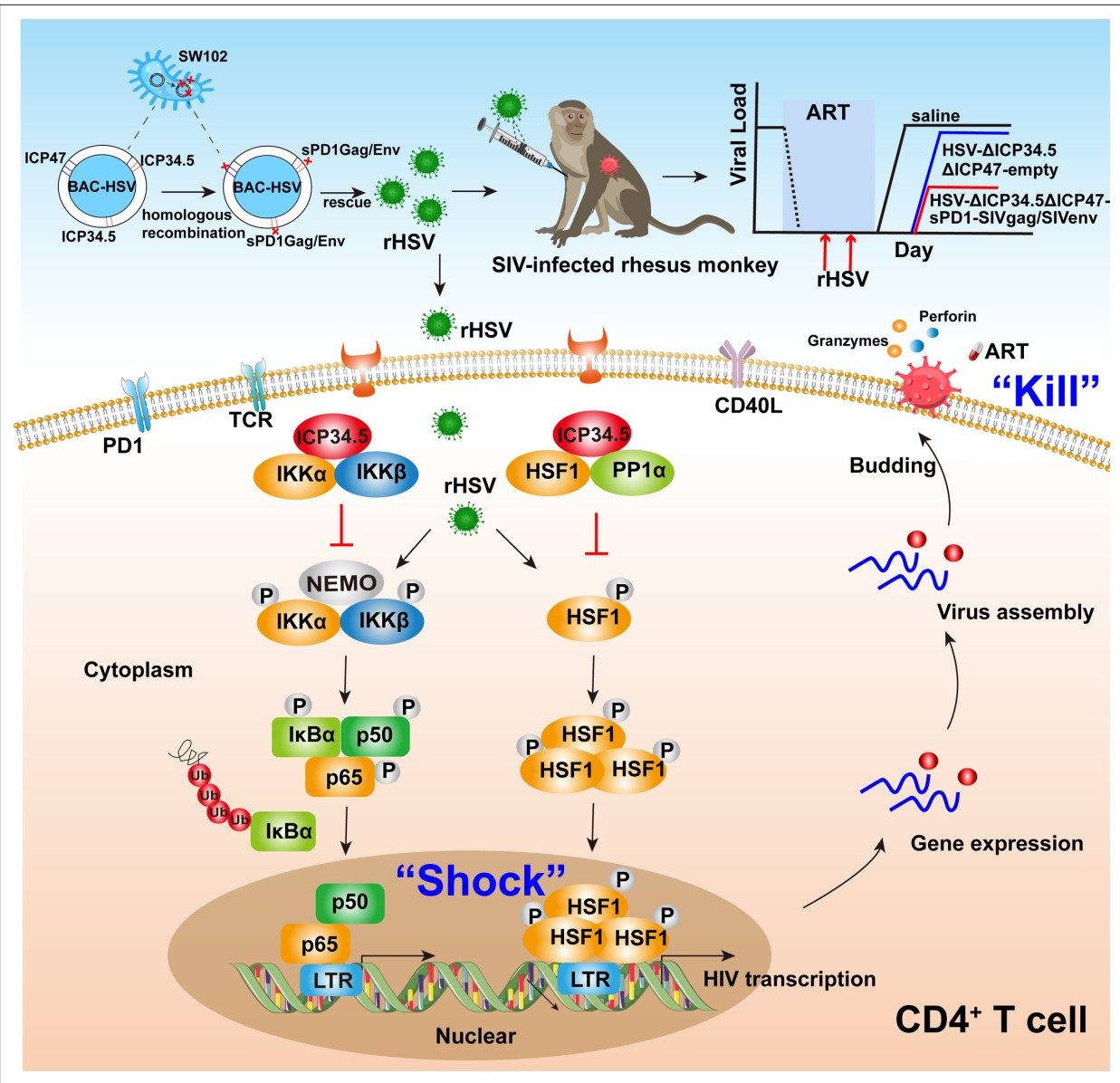

**Figure 6.** Pattern to illustrate the proof-of-concept strategy based on a bifunctional HSV-ΔICP34.5-vectored therapeutic vaccine for human immunodeficiency virus (HIV) functional cure. In the present study, the modified HSV-ΔICP34.5-based constructs effectively reactivated HIV/simian immunodeficiency virus (SIV) latency by modulating the IKKα/β-NF-κB pathway and PP1-HSF1 pathway (shock) and simultaneously elicited antigen-specific polyfunctional CD8+ T cells to eliminate cells infected with the reactivated virion (kill). BAC: bacterial artificial chromosome; rHSV: recombinant herpes simplex virus; TCR: T cell receptor; PD1: programmed cell death protein 1; CD40L: CD40 ligand.

been extensively investigated to target the latent HIV reservoirs for an HIV functional cure (*Deeks, 2012*; *Yeh and Ho, 2021*; *Maldini et al., 2020*; *Herzig et al., 2019*; *Dash et al., 2023*; *Dashti et al., 2023*; *Walker-Sperling et al., 2022*). However, there is no safe and effective approach for clinical use in HIV patients yet. In the present study, we occasionally found that the modified HSV-ΔICP34.5-based constructs could reactivate HIV latency more efficiently than wild-type HSV counterpart, which inspired us to develop a proof-of-concept strategy based on a bifunctional HSV-vectored therapeutic vaccine, aiming to simultaneously reactivate viral latency and elicit HIV/SIV-specific immune responses for HIV functional cure. Our results indicated that these modified HSV-based constructs efficiently elicited antigen-specific immune responses in mice and chronically SIV-infected macaques, and further therapeutic efficacy experiments showed that this strategy partly reactivated SIV latency in vivo and delayed viral rebound in chronically SIV-infected, ART-treated macaques (*Figure 6*).

The latent HIV proviruses can harbor it into the host genome with a quiescent transcription state, and thus cannot be recognized by immune surveillance or drug killing (*Chapman et al., 1991*; *Pierson et al., 2000*). Therefore, it is critical to disrupt viral latency for developing an HIV cure strategy. Based on our experimental data, the mechanism for efficiently reactivating viral latency by the modified HSV-ΔICP34.5-based constructs may involve regulating the IKKα/β-NF-κB pathway and PP1-HSF1 pathway. Indeed, during its replication, HSV can activate the double-stranded RNA-dependent protein kinase (PKR) pathway, and thus phosphorylate the protein translation initiation regulator eIF2α, resulting in the initiation of protein translation (*Farassati et al., 2001*). Previous studies have shown that the reactivation potential of HSV might be intertwined with NF-κB, Sp1, and other unknown transcription factors by ICP0, ICP4, and ICP27 (*Chapman et al., 1991*; *Amici et al., 2004*; *Vlach and Pitha, 1992*; *Mosca et al., 1987b*; *Golden et al., 1992*; *Vlach and Pitha, 1993*; *Mosca et al., 1987a*). However, the underlying mechanism by which the ICP34.5-deleted HSV construct can greatly improve the reactivation efficacy of HIV latency remains elusive.

ICP34.5 is a neurotoxicity factor that can antagonize innate immune responses, including PKR, TANK binding kinase (TBK1) signaling, and Beclin1-mediated apoptosis (*He et al., 1997*; *Manivanh et al., 2017*; *Orvedahl et al., 2007*). Thus, ICP34.5 deletion is beneficial to improve the safety of HSV-based constructs. As expected, our data have demonstrated experimentally that HSV-ΔICP34.5 exhibited lower virulence and replication ability than wild-type HSV-1 (*Figure 1—figure supplement 1*). Importantly, we also observed a significant decrease in the expression of inflammatory factors in PWLH when compared to wild-type HSV-1 (*Figure 1I–K*). These data suggested that the safety of HSV-ΔICP34.5 should be more tolerable than wild-type HSV-1. ICP34.5 binds to host PP1 and mediates the dephosphorylation of eIF2α, thus allowing protein synthesis and reversing the effects of PKR and host antiviral functions (*He et al., 1997*; *He et al., 1998*). In this study, our findings further unveiled an interaction between ICP34.5 and HSF1, resulting in reduced HSF1 phosphorylation via recruitment of PP1α. Interestingly, previous reports indicated that HSF1 could positively regulate HIV gene transcription (*Rawat and Mitra, 2011*), which is facilitated by its nuclear entry post-phosphorylation and subsequent recruitment of p300 for self-acetylation, along with binding to the HIV-1 LTR. Studies have also shown that HSF1 could further orchestrate p-TEFb recruitment to promote HIV-1 transcriptional elongation (*Peng et al., 2020*; *Lin et al., 2018*; *Pan et al., 2016b*; *Pan et al., 2016a*). Under stress-induced conditions, phosphorylation triggers the formation of the HSF1 trimer, thus facilitating its nuclear entry to bind to heat shock elements for gene transcription regulation (*Bonner et al., 2000*; *Timmons et al., 2020*). Additionally, we also demonstrated that ICP34.5 interacted with IKKα and IKKβ, thereby impeding NF-κB nuclear entry and further curbing HIV latency. Consistently, previous studies have also suggested that ICP34.5 could disrupt the NF-κB pathway and possibly affect the maturation of dendritic cells (*Jin et al., 2011*). Intriguingly, these findings collectively indicated that ICP34.5 might play an antagonistic role in the reactivation of HIV by HSV-1, and thus our modified HSV-ΔICP34.5 constructs can effectively reactivate HIV/SIV latency through the release of imprisonment from ICP34.5. However, ICP34.5 overexpression had only a partial effect on the reduction of the HIV latency reactivation, indicating that HSV-ΔICP34.5-based constructs can also reactivate HIV latency through other yet-to-be-determined mechanisms.

Another observation in this study is that the HSV-ΔICP34.5ΔICP47-sPD1-SIVgag/SIVenv construct elicited robust and persistent SIV-specific T cell immune responses in ART-treated, chronically SIV-infected macaques. Increasing evidence has indicated that HIV-specific CTLs can facilitate the suppression of latent viral reservoirs, and, thus, the induction of robust and persistent HIV-specific CTL

responses is essential for achieving long-term disease-free and transmission-free HIV control (**Collins et al., 2020**). Featured polyfunctional CD8+ T cells may contribute to HIV elite controllers or long-term non-progressors, a rare proportion of HIV-infected individuals who can spontaneously control viral replication even without ART treatment (**Owen et al., 2010**; **Ferre et al., 2009**; **O'Connell et al., 2009**). In addition, both our previous study and others have demonstrated that strong antigen-specific CD8+ T cell immune responses, especially effector memory CD8+ T cells, were associated with a lower VL, and in vivo CD8+ lymphocyte depletion with intravenous infusion of anti-CD8 monoclonal antibody could lead to dramatical viral rebound in these vaccinated elite macaques (**Perdomo-Celis et al., 2022**; **Pandrea et al., 2011**; **Sun et al., 2013**; **Sun et al., 2010**; **Pan et al., 2018**; **Sun et al., 2012**). Notably, previous studies have shown that HSV-based vaccines expressing HIV/SIV antigen did elicit specific CD8+ T cell immune responses in mice and macaques, but the magnitude was not as strong as other viral vectors (**Parker et al., 2007**; **Murphy et al., 2000**; **Kaur et al., 2007**). Therefore, to further improve its immunogenicity in vivo, some modifications were adopted to optimize the HSV-vectored vaccine. (1) The ICP47 protein, encoded by the US12 gene, can bind to the transporter associated with antigen presentation (TAP) 1/2, inhibiting the transport of viral peptides into the endoplasmic reticulum and the loading of peptides onto nascent major histocompatibility complex class I molecules to activate CD8+ T cells (**Orr et al., 2005**). Therefore, the ICP47 gene was deleted from our developed HSV vector. (2) Negatively immunoregulatory molecules, including PD1, TIM-3, and LAG-3, are usually involved in the pathogenesis of HIV infection, as well as in chronically SIV-infected macaques. Among them, PD1 upregulation can result in the exhaustion and dysfunction of CD8+ T cells (**Trautmann et al., 2006**). In addition, PD1 expression on memory CD4+ T cells might be linked with HIV latency. In this study, we modified the SIV antigen by fusing to sPD1, which can block the PD1/PDL1 pathway by competitively binding with PDL1 and thus improve HIV/SIV vaccine-induced CD8+ T cell immune responses (**Zhou et al., 2013**; **Wu et al., 2022**; **Pan et al., 2018**; **Xiao et al., 2014**).

Although promising, there are still some limitations to our study. First, this is a pilot study with a relatively small numbers of RMs, and future studies with a larger number of animals can be conducted to better verify our strategy. Second, the HSV-ΔICP34.5ΔICP47-sPD1-SIVgag/SIVenv vaccine resulted in delayed viral rebound and a lower peak VL post-rebound than pre-ART treatment but did not completely suppress SIV virus rebound in this study, which may be attributed to suboptimal doses and treatment, implying that we should further optimize this regimen for eventually achieving an HIV functional cure in future studies. The current consensus on HIV/AIDS vaccines emphasizes the importance of simultaneously inducing broadly neutralizing antibodies and cellular immune responses. Therefore, we believe that incorporating the induction of broadly neutralizing antibodies into our future optimizing approaches may lead to better therapeutic outcomes. Taken together, these findings demonstrated that our modified HSV-ΔICP34.5-based constructs potentially reactivated HIV/SIV latency by modulating the IKKα/β-NF-κB pathway and PP1-HSF1 pathway, and thereby we developed a proof-of-concept strategy based on a bifunctional HSV-vectored therapeutic vaccine, which can provide insights into the rational design of novel strategies for pursuing an HIV functional cure.

## Methods

### Mouse ethics statement and vaccination

Female BALB/c mice, aged 6–8 weeks, were procured from the Experimental Animal Center of Sun Yat-sen University. A total of 25 mice were randomly divided into five groups to evaluate the immunogenicity of the recombinant HSV-vector-based SIV vaccine. During the initial week, each mouse was subcutaneously administered a vaccination of $1 \times 10^6$ PFU of the respective recombinant HSV-vector vaccines (HSV-ΔICP34.5ΔICP47-empty, HSV-ΔICP34.5ΔICP47-sPD1, HSV-ΔICP34.5ΔICP47-SIVgag, HSV-ΔICP34.5ΔICP47-sPD1-SIVgag, HSV-ΔICP34.5ΔICP47-SIVenv). Following a 2-week interval, a booster vaccination was administered via the subcutaneous route using $2 \times 10^6$ PFU of recombinant HSV-vector vaccines. The subsequent assessment of the immune response involved ELISpot and ICS assays in accordance with the vaccination schedule.

### Ethics statement and vaccination of macaques

A total of nine chronically SIV$_{mac239}$-infected RMs were included in this study. All RMs received ART treatment (FTC/6.7 mg/kg/once daily, PMPA/10 mg/kg/once daily) for a duration of 33 days. The

RMs were allocated into three groups based on their baseline plasma SIV VL: ART+saline (n=3) as the control group, ART+HSV-ΔICP34.5ΔICP47-empty (n=3) as the sham group, and ART+HSV-ΔICP34.5ΔICP47-sPD1-SIVgag/SIVenv (n=3) as the vaccinated group. On day 0, all RMs received ART treatment. Once the VL dropped below the limit of detection ($1 \times 10^2$ copies/mL), the vaccinated group and sham group were primed with a subcutaneous vaccination of $1 \times 10^7$ PFU HSV-ΔICP34.5ΔICP47-sPD1-SIVgag/SIVenv or HSV-ΔICP34.5ΔICP47-empty, respectively, while the control group received an injection of 0.9% saline. After 3 weeks of the prime vaccination, the vaccinated group and sham group RMs received a subcutaneous booster vaccination of $5 \times 10^7$ PFU HSV-ΔICP34.5ΔICP47-sPD1-SIVgag/SIVenv or HSV-ΔICP34.5ΔICP47-empty, respectively. After 2 weeks of booster vaccination, the ART treatment for all RMs was interrupted. The immune response was evaluated by IFN-γ ELISpot and ICS assays. Plasma VL was regularly monitored throughout the studied period.

## Peptide pools

The peptide pools, encompassing the complete sequences of SIV Gag, Env, and Pol proteins, comprising 15 amino acids with 11 overlaps, were generously provided by the HIV Reagent Program, National Institutes of Health (NIH), USA. Gag pools comprise 125 peptides, while Env and Pol pools are subdivided into Env1 (109 peptides) and Env2 (109 peptides) pools, as well as Pol1 (131 peptides) and Pol2 (132 peptides) pools. These peptide pools were dissolved in dimethyl sulfoxide (DMSO, Sigma) at a concentration of 0.4 mg/peptide/mL for subsequent immunological assays.

## Construction of recombinant HSV

For the generation of recombinant HSV-vector-based vaccines, we implemented modifications at the ICP34.5 loci through homologous recombination within the BAC-HSV-1 system. Specifically, double copies of the ICP34.5 and ICP47 genes were either deleted or inserted into the respective counterparts: sPD1, SIVgag, sPD1-SIVgag, and SIVenv genes. The sPD1-SIVgag gene was created by fusing the N-terminal region of mouse sPD1 with the C-terminal segment of SIVgag, connected by a GGGSGGG linker, which was engineered through overlapping PCR.

## Plaque assay

Vero cells were seeded at a density of $2 \times 10^5$ cells per well in a 12-well plate and cultured for 24 hr. Virus samples were serially diluted and then added to the wells. After 2 hr of incubation at 37°C with 5% $CO_2$, the supernatant was aspirated, and the cells were washed with PBS. Then, a medium containing 1% fetal bovine serum (FBS), 1% low-melting-point agarose, and 1% penicillin-streptomycin was added to each well. After 3 days of incubation, cells were fixed with 4% paraformaldehyde at room temperature for 2 hr and stained with 1% crystal violet for 10 min. Plaques were visualized and counted, and virus titers were calculated as plaque-forming units per milliliter (PFU/mL).

## IFN-γ ELISpot assay

The IFN-γ ELISpot assay was conducted in accordance with our previous study (*Sun et al., 2010*). In the mouse experiment, $2.5 \times 10^5$ freshly isolated mouse spleen lymphocytes were simulated with SIV Gag, Env1, and Env2 peptide pools. In the monkey experiment, $2 \times 10^5$ PBMCs were seeded and simulated with the SIV Gag, Env, and Pol peptide pools. DMSO (Sigma) was utilized as Mock stimulation, while concanavalin A (ConA, 10 μg/mL) was employed as a positive control. Spot quantification was performed using an ELISpot reader (Mabtech), and peptide-specific spot counts were determined by subtracting the spots from mock stimulation.

## Intracellular cytokine staining

The ICS assay was performed as described previously (*Wu et al., 2022*). In the mouse experiment, $2 \times 10^6$ freshly isolated mouse spleen lymphocytes were stimulated with SIV Gag, Env1, and Env2 peptide pools. For the monkey experiment, $2 \times 10^6$ PBMCs were seeded and simulated with the SIV Gag, Env, and Pol peptide pools. DMSO and PMA (Thermo Scientific) plus ionomycin were used as the mock and positive controls, respectively. Data analysis was performed using FlowJo software (version 10.8.1), and the antibodies used are listed in *Supplementary file 1c*. The frequencies of cytokines produced from peptide-specific cells were analyzed by subtracting mock stimulation.

## SIV viral RNA and DNA copy assays and absolute T cell count

Absolute T cell count and the levels of plasma mRNA and SIV total DNA were quantified as described previously (*Wu et al., 2022*). Plasma viral RNA copy numbers were determined via SYBR green-based real-time quantitative PCR (Takara), using SIV gag-specific primers (*Supplementary file 1d*). Viral RNA copy numbers were calculated based on the standard curve established using SIV$_{mac239}$ gag standards. The lower limit of detection for this assay was 100 copies/mL of plasma. Total cellular DNA was extracted from approximately 0.5–5 million cells using a QIAamp DNA Blood Minikit (QIAGEN). PCR assays were performed with 200 ng samples of DNA, and SIV viral DNA was quantified using a pair of primers targeting a conserved region of the SIV gag gene, as previously described. Quantitation was performed by comparing the results to the standard curve of SIV gag copies. Integrated SIVmac239 DNA was measured using nested PCR with two sets of primers. The first round used SIVgag-reverse and Alu-forward primers, while the second round used primers specific to a conserved SIVgag region. Quantification was compared to a standard of CEMss/pWPXLD-rc cell genome as our previously described (*Yang et al., 2019*).

## Plasmids, cells, and viruses

pVAX-Myc-PDL1, pVAX-Flag-ICP34.5, pVAX-Myc-HSF1, pVAX-HA-PP1α: full-length mouse PDL1, full-length HSV-1 ICP34.5, and full-length human HSF1 and human PP1α with N- or C-terminal tag were cloned into the pVAX vector. pVAX-empty, pcDNA3.1-IKKα, and pcDNA3.1-IKKβ plasmids were stored in our laboratory. pVAX-empty was used as a mock transfection in our study.

293T cells (from the embryonic kidney of a female human fetus), Vero cells (from the kidney of a female normal adult African green monkey), and Hela cells (from the cervical cancer cells of an African American woman) were cultured in complete Dulbecco's modified Eagle's medium (Gibco) containing 10% FBS (Gibco) and 1% penicillin/streptomycin (Gibco) at 37°C in an atmosphere of 5% $CO_2$. The J-Lat 10.6 cells (Jurkat cells contain the HIV-1 full-length genome whose Env was frameshifted and inserted with GFP in place of Nef) and the HIV-1 latently infected CD4$^+$ CEM cells ACH-2 (A3.01 cell integrated HIV-1 proviral DNA) were cultured in complete RPIM640 (Gibco) containing 10% FBS and 1% penicillin/streptomycin at 37°C in an atmosphere of 5% $CO_2$. J-Lat 10.6- ICP34.5 cells were constructed in our laboratory. All cell lines used in this study were routinely tested for mycoplasma contamination using a PCR-based detection assay and confirmed to be mycoplasma-free.

The HSV-1 (McKrae) strains were stored in our laboratory. HSV-GFP (17 strain) was rescued from pBAC-GFP-HSV in Vero cells. HSV-ΔICP34.5 (17 strain) was rescued from pBAC-GFP-HSV-ΔICP34.5 (with deletion of double copies of ICP34.5 gene) in Vero cells.

## RT-qPCR

RT-qPCR was performed as described in our previous study (*Zhao et al., 2022*). Data were normalized to β-actin. The primer sequences are listed in *Supplementary file 1*. Fold changes in the threshold cycle (Ct) values were calculated using the $2^{-\Delta\Delta Ct}$ method.

## ChIP assay

ChIP analysis was conducted using the SimpleChIP Enzymatic Chromatin IP kit (Agarose Beads) (CST), following the manufacturer's instructions. Quantitative real-time PCR was employed for detecting the LTR sequence. The sequences of primers used in the LTR ChIP are listed in *Supplementary file 1*. % Input = 2% × $2^{[(ct)2\% \text{ input sample (ct) IP sample}]}$.

## Co-immunoprecipitation

Cells were harvested and subjected to Co-IP assay, following the protocol outlined in our previous study (*Zhao et al., 2022*).

## Protein extraction and western blot

Nuclear and cytoplasmic proteins were extracted by kits following the manufacturer's instructions (Beyotime). The western blotting assay was performed as previously described (*Zhao et al., 2022*).

## Statistical analysis

GraphPad Prism 8.0 (GraphPad Software, San Diego, CA, USA) was used for statistical analysis. For intragroup direct comparisons, Student's unpaired two-tailed t test was performed to analyze

significant differences. For comparisons of multiple groups, one-way ANOVAs were performed. Significance levels are indicated as *$p<0.05$, **$p<0.01$, ***$p<0.001$, ****$p<0.0001$.

## Study approval

Mice experiment was approved by the Laboratory Animal Ethics Committee guidelines at Sun Yat-sen University (approval number: SYSU-IACUC-2021-000185). Chinese RMs (*Macaca mulatta*) were housed at the Landau Animal Experimental Center, Guangdong Landau Biotechnology Co., Ltd. (approval number: N2021101). The primary PBMC samples were isolated from the chronically HIV-1-infected participants who were recruited from Guangzhou Eighth People's Hospital (Guangzhou, Guangdong, China), which was approved by the Medical Ethics Review Board of the School of Public Health (Shenzhen), Sun Yat-sen University (2022-037). All participants provided written informed consent, agreeing to participate in this study. The enrollment criteria for HIV-1-infected individuals included sustained suppression of plasma HIV-1 viremia under ART, with undetectable plasma HIV-1 RNA levels (less than 50 copies/mL) for at least 12 months and a $CD4^+$ T cell count more than 350 cells/μL.

## Acknowledgements

We appreciate the staff at the Animal Center of GIBH for their excellent technical assistance. We thank all other members of our group for their helpful advice and discussion to improve this project. We also appreciate the NIH AIDS Research and Reference Reagent Program for providing SIV peptide pools. This project was supported by National Natural Science Foundation of China (82472267, 82271786, 32370171), Science and Technology Planning Project of Guangdong Province, China (2021B1212040017), Shenzhen Key Laboratory of Pathogenic Microbes and Biosafety (ZDSYS20230626091203007), the Sanming Project of Medicine in Shenzhen Nanshan (SZSM202103008), and the Key Subject of Nanshan district of Shenzhen for AIDS surveillance and prevention.

## Additional information

### Funding

| Funder | Grant reference number | Author |
|---|---|---|
| National Nature Science Foundation of China | 82472267 | Caijun Sun |
| Science and Technology Planning Project of Guangdong Province | 2021B1212040017 | Caijun Sun |
| shenzhen key laboratory of pathogenic microbes and biosafety | ZDSYS20230626091203007 | Caijun Sun |
| the sanming project of medicine in shenzhen nanshan | SZSM202103008 | Caijun Sun |
| National Natural Science Foundation of China | 82271786 | Caijun Sun |
| National Natural Science Foundation of China | 32370171 | Caijun Sun |

The funders had no role in study design, data collection and interpretation, or the decision to submit the work for publication.

### Author contributions

Ziyu Wen, Data curation, Formal analysis, Investigation, Methodology, Writing – original draft, Writing – review and editing; Pingchao Li, Data curation, Methodology, Writing – review and editing; Yue Yuan, Data curation, Formal analysis, Methodology, Writing – review and editing; Congcong Wang, Data curation, Investigation, Methodology, Writing – review and editing; Minchao Li, Haohang Wang,

Minjuan Shi, Yizi He, Mingting Cui, Methodology; Ling Chen, Resources, Project administration, Writing – review and editing; Caijun Sun, Conceptualization, Resources, Supervision, Funding acquisition, Writing – original draft, Project administration, Writing – review and editing

**Author ORCIDs**
Ziyu Wen (iD) https://orcid.org/0009-0004-2552-4377
Caijun Sun (iD) https://orcid.org/0000-0002-2000-7053

**Ethics**
The primary PBMC samples were isolated from the chronically HIV-1-infected participants who were recruited from Guangzhou Eighth People's Hospital (Guangzhou, Guangdong, China), which was approved by the Medical Ethics Review Board of the School of Public Health (Shenzhen), Sun Yat-sen University (2022-037). All participants provided written informed consent, agreeing to participate in this study.
Mice experiment was approved by the Laboratory Animal Ethics Committee guidelines at Sun Yat-sen University (approval number: SYSU-IACUC-2021-000185).

Reviewer #1 (Public review): https://doi.org/10.7554/eLife.95964.4.sa1
Author response https://doi.org/10.7554/eLife.95964.4.sa2

# Additional files

**Supplementary files**
Supplementary file 1. Supplementary figures and tables for this study. (a) Immunoprecipitation-mass spectrometry (IP-MS) analysis. (b) Baseline information for the experimental macaques in this study. (c) Antibodies for the intracellular cytokine staining (ICS) assay in this study. (d) The sequences of primers.

MDAR checklist

**Data availability**
All data generated or analysed during this study are included in the manuscript and supporting files.

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
