## [Editor Report · eLife Assessment]

In this **useful** study, the authors tested a novel approach to eradicate the HIV reservoir by constructing a herpes simplex virus (HSV)-based therapeutic vaccine designed to reactivate HIV from latently infected cells and induce an immune response to kill such infected cells. Testing this approach with SIV in a primate model, the authors report that the SIV reservoir was reduced. However, the evidence presented appears to be **incomplete** because the animal group size was small and the SIV reservoir size highly variable.

---

## [Referee Report · Reviewer #1 (Public review)]

Summary:

Authors constructed a novel HSV-based therapeutic vaccine to cure SIV in a primate model. The novel HSV vector is deleted for ICP34.5. Evidence is given that this protein blocks HIV reactivation by interference with the NFkappaB pathway. The deleted construct supposedly would reactivate SIV from latency. The SIV genes carried by the vector ought to elicit a strong immune response. Together the HSV vector would elicit a shock and kill effect. This is tested in a primate model.

Strengths and weaknesses:

(1) Deleting ICP34.5 from the HSV construct has a very strong effect on HIV reactivation. The mechanism underlying increased activation by deleting ICP34.5 is only partially explored. Overexpression of ICP34.5 has a much smaller effect (reduction in reactivation) than deletion of ICP34.5 (strong activation); this is acknowledged by the authors that no full mechanistic explanation can be given at this moment.

(2) No toxicity data are given for deleting ICP34.5. How specific is the effect for HIV reactivation? A RNA seq analysis is required to show the effect on cellular genes.

A RNA seq analysis was done in the revised manuscript comparing the effect of HSV-1 and deleted vector in J-LAT cells (Fig S5). More than 2000 genes are upregulated after transduction with the modified vector in comparison with the WT vector. Hence, the specificity of upregulation of SIV genes is questioned. Authors do NOT comment on these findings. In my view it questions the utility of this approach.

(3) The primate groups are too small and the results to variable to make averages. In Fig 5, the group with ART and saline has two slow rebounders. It is not correct to average those with the single quick rebounder. Here the interpretation is NOT supported by the data.

Although authors provided some promising SIV DNA data, no additional animals were added. Groups of 3 animals are too small to make any conclusion, especially since the huge variability in response. The average numbers out of 3 are still presented in the paper, which is not proper science.

No data are given of the effect of the deletion in primates. Now the deleted construct is compared with an empty vector containing no SIV genes. Authors provide new data in Fig S2 on the comparison of WT and modified vector in cells from PLWH, but data are not that convincing. A significant difference in reactivation is seen for LTR in only 2/4 donors and in Gag in 3/4 donors. (Additional question what is meaning of LTR mRNA, do authors relate to genomic RNA??)

Discussion

HSV vectors are mainly used in cancer treatment partially due to induced inflammation. Whether these are suitable to cure PLWH without major symptoms is a bit questionable to me and should at least be argued for.

The RNA seq data add on to this worry and should at least be discussed.

Comments on revisions:

The authors accept the limitations of the primate study (too small for strong conclusions). The new way of presenting the data clearly shows these limitations.

---

## [Author Response]

The following is the authors’ response to the previous reviews.

**Public Reviews:**

**Reviewer #1 (Public review):**
Summary:The authors constructed a novel HSV-based therapeutic vaccine to cure SIV in a primate model. The novel HSV vector is deleted for ICP34.5. Evidence is given that this protein blocks HIV reactivation by interference with the NFkappaB pathway. The deleted construct supposedly would reactivate SIV from latency. The SIV genes carried by the vector ought to elicit a strong immune response. Together the HSV vector would elicit a shock and kill effect. This is tested in a primate model.Strengths and weaknesses:(1) Deleting ICP34.5 from the HSV construct has a very strong effect on HIV reactivation. The mechanism underlying increased activation by deleting ICP34.5 is only partially explored. Overexpression of ICP34.5 has a much smaller effect (reduction in reactivation) than deletion of ICP34.5 (strong activation); this is acknowledged by the authors that no full mechanistic explanation can be given at this moment.

Thank you for your comments. We agree with you that the mechanism underlying increased reactivation by deleting ICP34.5 is only partially explored. As you pointed out, the deletion of ICP34.5 leads to a significant reactivation, while the overexpression of ICP34.5 has a relatively weak inhibitory effect on reactivation. This difference prompts us to further contemplate the role of HSV-1 in regulating HIV latency and reactivation. Our data (Figure S4), along with previous literature (Mosca et al., 1987, Nabel et al., 1988), have indicated that the ICP0 protein might play a crucial role in the reactivation of HIV latency. However, we found for the first time that ICP34.5 can play an antagonistic role with this reactivation. This is a very interesting topic for understanding the complicated interactions between host cells and different viruses. We will investigate the deeper insights in future studies, and we have mentioned this limitation in the revised Discussion Section. Thank you!

(2) No toxicity data are given for deleting ICP34.5. How specific is the effect for HIV reactivation? A RNA seq analysis is required to show the effect on cellular genes.A RNA seq analysis was done in the revised manuscript comparing the effect of HSV-1 and deleted vector in J-LAT cells (Fig S5). More than 2000 genes are upregulated after transduction with the modified vector in comparison with the WT vector. Hence, the specificity of upregulation of SIV genes is questioned. Authors do NOT comment on these findings. In my view it questions the utility of this approach.

Thank you for your mentions.

(1) As for the toxicity of HSV-ΔICP34.5, it is well known that ICP34.5 is a neurotoxicity factor that can antagonize host immune responses, and thus deleting ICP34.5 is beneficial to improve the safety of HSV-based constructs. As expected, we have demonstrated experimentally that HSV-DICP34.5 exhibited lower virulence and replication ability than wild-type HSV-1 (Figure S1). Importantly, we also observed a significant decrease in the expression of inflammatory factors in PWLH when compared to wild-type HSV-1 (Figure 1I-K). These data suggested that the safety of HSV-DICP34.5 should be more tolerable than wild-type HSV vector.

(2) The RNASeq analysis is aimed to explore the HSV-ΔICP34.5-induced signaling pathways, but it is not suitable to use this data for assessing the toxicity of HSV-ΔICP34.5 constructs. As for the RNASeq data, we think it is reasonable to observe many upregulated genes (which are involved in a variety of signaling pathways), since HSV-DICP34.5 constructs reactivated HIV latency more effectively than wild-type HSV by modulating the IKKα/β-NF-kB pathway and PP1-HSF1 pathway.

(3) To further validate whether HSV-ΔICP34.5 can specifically activate the HIV latent reservoir, we conducted additional experiments using vaccinia virus and adenovirus as controls, and results showed that both vaccinia virus and adenovirus cannot effectively reactivate HIV latency (Figure S3). Moreover, the deletion of ICP0 gene from HSV-1 diminished the reactivation effect of HIV latency by HSV-1, and overexpressing ICP0 greatly reactivate the latent HIV (Figure S4, Figure S5), implying that this reactivation should be virus-specific and ICP0 plays an important factor on reversing HIV latency. Interestingly, we herein found that ICP34.5 can act as an antagonistic factor for this reactivation of HIV latency by HSV-1. Thus, after the deletion of ICP34.5, the ability of HSV to reverse HIV latency was significantly enhanced. Our research group will investigate the underlying mechanism in future studies. Thank you for your insightful mention.

(3) The primate groups are too small and the results to variable to make averages. In Fig 5, the group with ART and saline has two slow rebounders. It is not correct to average those with the single quick rebounder. Here the interpretation is NOT supported by the data.Although authors provided some promising SIV DNA data, no additional animals were added. Groups of 3 animals are too small to make any conclusion, especially since the huge variability in response. The average numbers out of 3 are still presented in the paper, which is not proper science.No data are given of the effect of the deletion in primates. Now the deleted construct is compared with an empty vector containing no SIV genes. Authors provide new data in Fig S2 on the comparison of WT and modified vector in cells from PLWH, but data are not that convincing. A significant difference in reactivation is seen for LTR in only 2/4 donors and in Gag in 3/4 donors. (Additional question what is meaning of LTR mRNA, do authors relate to genomic RNA??)

Thank you for your serious review and kind reminder.

(1) We agree with you that it is not appropriated to use averages for this pilot study with limited numbers of macaques. We are currently unable to conduct another experiment with a larger number of macaques, but we think the results of this pilot study were very promising for further studies. Now, following your kind suggestions, we have removed the averages and now presented the data for each monkey individually in the revised manuscript. We have also modified the corresponding description accordingly (Line 254 to 262). Thank you for your understanding.

(2) Regarding your comment about the lack of data on the deletion of ICP34.5 from HSV-1, we are sorry for previously unclear description. In fact, the empty vector used in our animal experiments not only does not contain SIV antigens but also has the ICP34.5 deletion. We have revised the corresponding description accordingly (For example, we use HSV-DICP34.5DICP47-empty, HSV-DICP34.5DICP47-sPD1-SIVgag/SIVenv instead of HSV-empty, HSV-sPD1-SIVgag/SIVenv). We hope this revision will address your question.

(3) As for the reactivation effects observed in PLWH samples, the data may be not perfect, but we think this result (a significant difference in reactivation is seen for LTR in 2/4 donors and for Gag in 3/4 donors, and the purpose of detecting LTR RNA is to evaluate the level of virus replication) is promising to support our conclusion (The enhanced reactivation effect in primary CD4+ T cells by HSV-∆ICP34.5 than wild-type HSV). Of course, we recognize the need for more samples to gain a comprehensive understanding of reactivation effect in different individuals in future study. In addition, we corrected the description of LTR RNA (Lines 99-106 and 115-116). Thank you for the reminder!

DiscussionHSV vectors are mainly used in cancer treatment partially due to induced inflammation. Whether these are suitable to cure PLWH without major symptoms is a bit questionable to me and should at least be argued for.The RNA seq data add on to this worry and should at least be discussed.

Thank you for your mention. As mentioned above, the RNASeq analysis is aimed to explore the HSV-ΔICP34.5-induced signaling pathways, but it is not suitable to use this data for assessing the toxicity of HSV-ΔICP34.5 constructs. Actually, ICP34.5 is a neurotoxicity factor that can antagonize innate immune responses, and thus ICP34.5 deletion is beneficial to improve the safety of HSV-based constructs. As expected, our data have demonstrated experimentally that HSV-DICP34.5 exhibited lower virulence and replication ability than wild-type HSV-1 (Figure S1). Importantly, HSV-DICP34.5 induced a lower level of inflammatory cytokines (including IL-6, IL-1β, and TNF-α) in primary CD4+ T cells from PLWH compared to HSV stimulation, likely due to its lower virulence and replication ability (Figure 1I-K). In addition, the CD4+ /CD8+ T cell ratio (Figure 5H) and body weight (Figure S10) after treatment were effectively ameliorated in the SIV-infected macaques of the ART+HSV-DICP34.5DICP47-sPD1-SIVgag/SIVenv group. Our data also demonstrated that there was no significant effect on the cell composition of peripheral blood in the SIV-infected macaques of ART+HSV-DICP34.5DICP47-sPD1-SIVgag/SIVenv group (Figure S11). These data suggested that the safety of HSV-DICP34.5 should be more tolerable than wild-type HSV vector. We have added a more comprehensive description in the revised Discussion (Lines 328-334). Thank you again for all of your kind comments and suggestions.

**Reviewer #2 (Public review):**
Summary:In this article Wen et. al., describe the development of a 'proof-of-concept' bi-functional vector based out of HSV-deltaICP-34.5's ability to purge latent HIV-1 and SIV genomes from cells. They show that co-infection of latent J-lat T-cell lines with a HSV-deltaICP-34.5 vector can reactivate HIV-1 from a latent state. Over- or stable expression of ICP 34.5 ORF in these cells can arrest latent HIV-1 genomes from transcription, even in the presence of latency reversal agents. ICP34.5 can co-IP with- and de-phosphorylate IKKa/b to block its interaction with NF-k/B transcription factor. Additionally, ICP34.5 can interact with HSF1 which was identified by mass-spec. Thus, the authors propose that the latency reversal effect of HSV-deltaICP-34.5 in co-infected JLat cells is due to modulatory effects on the IKKa/b-NF-kB and PP1-HSF-1 pathway.Next the authors cleverly construct a bifunctional HSV based vector with deleted ICP34.5 and 47 ORFs to purge latency and avoid immunological refluxes, and additionally expand the application of this construct as a vaccine by introducing SIV genes. They use this 'vaccine' in mouse models and show the expected SIV-immune responses. Experiments in rhesus macaques (RM), further elicit potential for their approach to reactivate SIV genomes and at the same time block their replication by antibodies. What was interesting in the SIV experiments is that the dual-functional vector vaccine containing sPD1- and SIV Gag/Env ORFs effectively delayed SIV rebound in RMs and in some cases almost neutralized viral DNA copy detection in serum. Very promising indeed, however there are some questions I wish the authors explored to answer, detailed below.Overall, this is an elegant and timely work demonstrating the feasibility of reducing virus rebound in animals, and potentially expand to clinical studies. The work was well written, and sections were clearly discussed.Strengths:The work is well designed, rationale explained and written very clearly for lay readers.Claims are adequately supported by evidence and well designed experiments including controls.

We appreciate your positive comment for our work.

Weaknesses:(1) It looks like ICP0 is also involved in latency reversal effects. More follow-up work will be required to test if this is in fact true.

Both our data (Figure S4, Figure S5) and previous literature (Nabel et al., 1988, Mosca et al., 1987) have reported that HSV ICP0 may play a role in reversing HIV latency. However, the exact mechanisms behind this effect have not yet been fully elucidated. Of note, we herein reported for the first time that ICP34.5 can act as an antagonistic factor for this reactivation of HIV latency by HSV-1. Thus, after the deletion of ICP34.5, the ability of HSV to reverse HIV latency was significantly enhanced. Our research group will investigate the underlying mechanism in future studies. Thank you for your insightful mention.

(2) It is difficult to estimate the depletion of the latent viral reservoir. The authors have tried to address this issue. A more convincing argument to this reviewer will be data to demonstrate that after the bi-functional vaccine, the animals show overall reduction in the number of circulating latent cells. The feasibility to obtain such a result is not clearly demonstrated.

Thank you for your comment. As you mentioned, we have indeed measured both total DNA and integrated DNA (iDNA) in blood cells (see Figure 5E-F), which can provide support for the reduction of the latent viral reservoir. Thank you for your kind reminder.

(3) The authors state that the reduced virus rebound detected following bi-functional vaccine delivery is due to latent genomes becoming activated and steady-state neutralization of these viruses by antibody response. This needs to be demonstrated. Perhaps cell-culture experiments from specimen taken from animals might help address this issue. In lab cultures one could create environments without antibody responses, under these conditions one would expect higher level of viral loads being released in response to the vaccine in question.

Thank you for your valuable suggestion. We believe that the reduced virus rebound observed may be influenced by immune responses from T cells and antibodies induced by both ART and the vaccine. We appreciate your insight and agree that future studies should focus on investigating the activation effects of the vaccine under controlled conditions that simulate the absence of immune responses in primary animal cells. This will help us better understand the mechanisms involved and address your concerns more comprehensively.

**Reviewer #2 (Recommendations for the authors):**
The Authors have sufficiently addressed my comments. Below are a few minor changes that can help with clarity.Lines 126-127: This sentence should be changed. Perhaps, "these data suggests that .... Safety of... in PLWH might be tolerable, at least in vitro."

Thanks for your suggestion. We have revised it accordingly. (Line 130).

Lines 128-132: Would this not mean that reactivation is due to ICP0 gene? Have the authors tried to express ICP0-gene into J-Lat cells and see if that is the reason for reactivation? This seems somewhat incomplete. At the end of 132, please add ", in the presence of ICP0". Also a sentence describing this effect is warranted.

Thank you for your insightful suggestion. Yes, both our data and previous literature supported that the ICP0 gene can play a significant role in the reactivation of HIV latency (Figure S4, Figure S5). Of note, we herein reported for the first time that ICP34.5 can act as an antagonistic factor for this reactivation of HIV latency by HSV-1. Thus, after the deletion of ICP34.5, the ability of HSV to reverse HIV latency was significantly enhanced. We have described this effect in the revised version accordingly. Additionally, we have added the phrase “in the presence of ICP0” to the results section (Lines 137) to clarify this point.

MOSCA, J. D., BEDNARIK, D. P., RAJ, N. B., ROSEN, C. A., SODROSKI, J. G., HASELTINE, W. A., HAYWARD, G. S. & PITHA, P. M. 1987. Activation of human immunodeficiency virus by herpesvirus infection: identification of a region within the long terminal repeat that responds to a trans-acting factor encoded by herpes simplex virus 1. *Proc Natl Acad Sci U S A* 84: 7408.DOI: https://doi.org/10.1073/pnas.84.21.7408, PMID: 2823260

NABEL, G. J., RICE, S. A., KNIPE, D. M. & BALTIMORE, D. 1988. Alternative mechanisms for activation of human immunodeficiency virus enhancer in T cells. *Science* 239: 1299.DOI: https://doi.org/10.1126/science.2830675, PMID: 2830675